# The Use of Natural Bioactive Nutraceuticals in the Management of Tick-Borne Illnesses

**DOI:** 10.3390/microorganisms11071759

**Published:** 2023-07-05

**Authors:** Samuel M. Shor, Sunjya K. Schweig

**Affiliations:** 1Internal Medicine of Northern Virginia, George Washington University Health Care Sciences, Reston, VA 20190, USA; 2California Center for Functional Medicine, Oakland, CA 94619, USA; sunjya@ccfmed.com

**Keywords:** *Babesia*, babesiosis, *Bartonella*, biofilm, borreliosis, *Borrelia burgdorferi*, *B. burgdorferi*, chronic Lyme disease, Lyme disease, persister cells, botanical medicine, herbal medicine, micronutrients, nutraceuticals, phytochemicals

## Abstract

The primary objective of this paper is to provide an evidence-based update of the literature on the use of bioactive phytochemicals, nutraceuticals, and micronutrients (dietary supplements that provide health benefits beyond their nutritional value) in the management of persistent cases of *Borrelia burgdorferi* infection (Lyme disease) and two other tick-borne pathogens, *Babesia* and *Bartonella* species. Recent studies have advanced our understanding of the pathophysiology and mechanisms of persistent infections. These advances have increasingly enabled clinicians and patients to utilize a wider set of options to manage these frequently disabling conditions. This broader toolkit holds the promise of simultaneously improving treatment outcomes and helping to decrease our reliance on the long-term use of pharmaceutical antimicrobials and antibiotics in the treatment of tick-borne pathogens such as *Borrelia burgdorferi*, *Babesia*, and *Bartonella*.

## 1. Introduction

Lyme disease results from an active infection from any of several pathogenic members of the *Borrelia burgdorferi sensu lato* complex (*Bbsl*), including *Borrelia burgdorferi sensu stricto (Bbss)* and *Borrelia mayonii* (in North America) and *Borrelia burgdorferi sensu strictu*, *Borrelia afzelii*, and *Borrelia garinii* (in Eurasia). Lyme disease is the most common vector-borne illness in the United States [1] and Europe [2], and the Centers for Disease Control and Prevention (CDC) estimates that the annual incidence of Lyme disease in the US is at least 476,000 [3,4]. 

While a standard course of antibiotics may effectively treat many people with acute Lyme disease, there is mounting evidence that a typical course of antibiotics for acute Lyme disease may leave up to 15–35% of people with chronic symptoms and health issues [5]. While the exact reasons for this ongoing illness remain an area of active research and discussion, various investigators point to persistent infection as a possible and important cause. In categorizing chronic active Lyme disease (CLD) cases, a consensus publication described two categories. The first category, CLD-U, describes an untreated group with clinical features consistent with Lyme disease, likely representing active infection, that for various reasons was never identified and/or treated. The second category, CLD-PT, or the “previously treated” group, represents those appropriately identified, but for which the treatment was inadequate. In this group, suspected active infection was thought to explain ongoing clinical features. Both groups were symptomatic for at least 6 months [6].

The pathophysiology of chronic active (CLD-PT) *Bburgdorferi* infection [6] is supported by both animal and human persistence studies [7,8,9,10,11,12,13,14,15,16,17,18,19,20,21,22,23,24,25,26,27,28,29,30,31,32,33]. Studies have shown that *B. burgdorferi* species can exist in three morphological forms. The typical motile corkscrew spirochete represents the vegetative (active) form. Under stress conditions, including changes in pH, osmolarity, and antibiotic exposure in vitro, the motile spirochetes can transform into latent persistent stationary forms and/or biofilm-like aggregates [34,35,36,37,38,39,40,41]. Additional survival mechanisms of *B. burgdorferi* persistence include “immune evasion via physical seclusion within immunologically protected tissue sites such as the CNS, joints and eyes, collagen-rich tissues, cells and biofilms; alterations in outer surface protein (OSP) profiles through antigenic variation, phasic variations, immune modulation via alterations in complement, neutrophil and dendritic cell functioning, and changes in cytokine and chemokine levels” [6].

Biofilm generation by pathogenic bacteria, including *B. burgdorferi* species [36,42], is another well-known mechanism of microbe persistence which can potentially lead to CLD. The presence of *B. afzelii* embedded in biofilm has been demonstrated in infected human tissues [42]. Of those diagnosed with tick-borne diseases, growing numbers are being found to have polymicrobial infections [43,44,45,46]. One such pathogen is the malaria-like intraerythrocytic parasite *Babesia* [47,48,49,50]. Research by Dunn and colleagues [48] suggests that the concomitant presence of *B. burgdorferi* in ticks contributes to the emergence and geographic expansion of *Babesia microti*. Further, those infected with multiple tick-borne pathogens tend to have more severe courses [51], contributing to increased challenges in achieving recovery.

The cornerstone of therapy for tick-borne infections has been antimicrobials. The value of using prolonged antibiotic courses is controversial [52,53,54], with differing recommendations. Guidelines and clinicians recommending shorter durations of treatment cite a perceived lack of therapeutic benefit. This impression is largely based on their interpretation of the findings of four NIH-funded trials [55,56,57]. However, the first two studies [55] were felt to have significant design and methodology flaws, resulting in questions as to their generalizability [58,59]. Importantly, the other two studies had sub-cohort analysis supporting the potential value of longer duration antimicrobial therapy [56,57]. 

Potential *B. burgdorferi* persistence causing chronic infection affirms the need for longer duration treatment than the current IDSA recommendations [54,58,60]. In our opinion, the clinical course and required treatment duration are dependent upon many factors. These include the severity of the individual case, the duration of illness prior to treatment, comorbidities, and the presence of coinfections or opportunistic infections [43,44]. Rather than using an arbitrary, “one size fits all” guideline [52,53], analysis and treatment decisions are made by the clinician at the point of care, using their clinical expertise and judgement [54]. 

Precedence exists for the appropriateness and necessity of prolonged courses of antimicrobials in clinical practice. Examples include multidrug-resistant tuberculosis, where the recommended treatment duration can be up to 15–24 months after culture conversion [61]. Additional examples include histoid leprosy, caused by *Mycobacterium leprae*, which requires 12–24 months of therapy [62], and Q fever endocarditis caused by the organism *Coxiella burnetti*, where one study showed that most patients required 18–24 months of treatment to sterilize their infected heart valves [63]. 

Due to the frequent polymicrobial and persistent nature of many tick-borne infections, treatment often relies upon prolonged duration and multidrug protocols [64]. However, it is critical to balance optimal antibiotic stewardship and the goal of minimizing the overuse of antibiotics, with the potential risk of driving antibiotic resistance. By understanding the pathophysiology of persistent Lyme disease and other tick-borne illnesses and using botanical medicines and nutraceuticals to target pleomorphic persister cells and biofilms [33,34,35,36,37,38,39,40,41,42], we have the potential to be more judicious in the use of prescription antibiotics. In so doing, we also have the greater potential to decrease the duration and complexity of antibiotic protocols, while improving outcomes. In addition, botanical medicines and nutraceuticals can have antimicrobial efficacy and potency against persister forms and thus have the potential to treat tick-borne illnesses independently of prescription antibiotics.

## 2. Methods

Using PUB MED, comprehensive searches were employed including using the following search terms, “Lyme disease”, “*Borrelia*”, “*Bartonella*”, “*Babesia*”, and cross referencing with generic concepts such as “herbals”, “nutraceuticals”, and “botanical medicines” along with each of the multiple agents described in this manuscript. Over 400 references were identified and reviewed. To meet inclusion criteria, studies had to include agents shown to have a therapeutic effect on at least one of the following: *Borrelia burgdorferi* and/or *Bartonella* species in active, stationary or biofilm forms, or activity against *Babesia* strains. Papers not demonstrating treatment evidence against any of these pathogens were excluded and thus not referenced.

## 3. Discussion

Botanical medicines and natural antimicrobial agents have been used for thousands of years and have been shown to be effective against various pathogens [65]. Goc and colleagues [66] tested 15 phytochemicals and micronutrients for activity against three morphological forms of *Borrelia burgdorferi* and *Borrelia garinii* (spirochetes, latent rounded forms, and biofilm). Their results showed that the most potent substances against the spirochete and rounded forms of *B. burgdorferi* and *B. garinii* were cis-2-decenoic acid, baicalein, monolaurin, and kelp (iodine), and that only baicalein and monolaurin revealed significant activity against the biofilm form [66]. Table 1 reflects a summary of organic oils studied by Goc and colleagues [67].

Feng and colleagues [68] investigated numerous agents for potential in vitro anti-*Borrelia burgdorferi* activity. Among these natural products, seven were found to have good activity against the stationary phase *B. burgdorferi* culture compared to the control antibiotics doxycycline and cefuroxime. The identified botanicals included *Cryptolepis sanguinolenta*, *Juglans nigra* (Black walnut), *Polygonum cuspidatum* (Japanese knotweed), *Artemisia annua* (Sweet wormwood), *Uncaria tomentosa* (Cat’s claw), *Cistus incanus*, and *Scutellaria baicalensis* (Chinese skullcap) [68].

Some essential oils extracted from plants have also been reported in the literature to have antimicrobial activities [69,70,71,72]. In a more recent in vitro essential oil study, Feng and colleagues [73] evaluated the activity of 34 essential oils in a *B. burgdorferi* stationary phase culture persister model. In this study, the top five essential oils (oregano, cinnamon bark, clove bud, citronella, and wintergreen) showed high anti-persister activity. Importantly, some of the essential oils were found to have excellent anti-biofilm ability, as shown by their ability to dissolve the aggregated biofilm-like structures. The top three hits, oregano, cinnamon bark, and clove bud, completely eradicated all viable cells without any regrowth in subcultures in fresh medium.

Furthermore, a number of researchers have explored the utility of botanical medicines for the treatment of *Babesia* infections [74,75,76,77]. According to Zhang e and colleague [78], some botanical medicines and their active constituents have potent activity against *B. duncani* in vitro and may be further explored for the more effective treatment of babesiosis. In a recent study that used a hamster erythrocyte model, a panel of herbal medicines was screened and *Cryptolepis sanguinolenta*, *Artemisia annua*, *Scutellaria baicalensis*, *Alchornea cordifolia*, and *Polygonum cuspidatum* were identified to have good in vitro inhibitory activity against *B. duncani* [74]. In another study, Zhang and colleagues [77] described how garlic oil and black pepper oil were active against *B. duncani.* Furthermore, Batiha and colleagues [76] found that *C. verum* extracts were potential anti-piroplasm drugs through in vitro and animal in vivo models. 

*Bartonella* infection had previously been thought to only be responsible for an acute, self-limiting illness. One such example is cat-scratch disease. Caused by *Bartonella henselae* transmitted during a cat bite or scratch, this can frequently manifest as an acute self-limited febrile illness associated with regional lymphadenopathy. However, evidence is now supporting a more complex perspective where subsets of infected, immunocompetent patients can become chronically bacteremic [78,79,80]. There is no single treatment effective for systemic *B. henselae* infection, and antibiotic therapy exhibited poor activity against typical uncomplicated cat scratch disease [81]. With possible persistent infection, *Bartonella* spp. infections can represent a significant treatment challenge [81]. Similar to *Borrelia*, stationary phase and biofilm persister cells have been identified with *Bartonella henselae* [82,83,84,85]. Standard treatments have a relatively poor impact on persister cells, and this is likely one important factor contributing to treatment failure and the persistence of infection [82]. 

Creative approaches to address mechanisms of persistence have been studied. Li and colleagues [84] worked to identify FDA-approved drugs with activity against stationary phase *Bartonella henselae*. In their study, they were able to show that pyrantel pamoate, daptomycin, methylene blue, clotrimazole, gentamicin, and streptomycin, at their respective maximum serum drug concentration (C_max_), had the capacity to completely eradicate stationary phase *B. henselae* after 3 days’ drug exposure in subculture studies. In this study, while the currently used drugs for treating bartonellosis, including rifampin, erythromycin, azithromycin, doxycycline, and ciprofloxacin, had very low minimal inhibitory concentration (MIC) against growing *B. henselae*, they had relatively poor activity against stationary phase *B. henselae* [84]. Given these findings, there has been growing interest in identifying additional active agents to treat *Bartonella* infections and the use of phytochemicals has been increasingly explored [82,84,85]. For example, two active ingredients of the essential oils oregano and cinnamon bark, carvacrol and cinnamaldehyde, respectively, were shown to be highly active against the stationary phase *B. henselae* and able to eradicate all the bacterial cells [82]. 

Appendix A provides an overview of many of the nutraceuticals studied. Included in this summary are agents identified as the most effective agents in a particular study. The key to these data with appropriate references can be reviewed in Appendix B.

### 3.1. Specific Agents

#### 3.1.1. *Alchornea cordifolia* Extracts

*Alchornea cordifolia* has documented antimicrobial and anti-inflammatory activity [86,87]. In addition, *A. cordifolia* has significant antimalarial effects [88,89,90] and has been used by traditional herbalists for the treatment of malaria in several African countries [91]. According to Zhang and colleagues [74], this botanical medicine showed good inhibitory effects against *Babesia duncani*. Importantly, preclinical studies revealed that *Alchornea cordifolia* has favorable toxicology and bioavailability profiles [91,92,93,94]. 

#### 3.1.2. Allicin (Garlic)

Allicin represents a key component of garlic, with a broad spectrum of antimicrobial activities [95]. These actions include (i) a therapeutic effect against a wide range of Gram-negative and Gram-positive bacteria; (ii) antifungal activity, such as *Candida albicans*; (iii) antiparasitic activity, such as *Entamoeba histolytica* and *Giardia lamblia*; and (iv) antiviral activity [95]. According to Feng, when assessing the activity of 35 essential oils in eradicating *B. burgdorferi* stationary forms in vitro, “garlic essential oil exhibited the best activity as shown by the lowest residual viability of *B. burgdorferi* [68,96]”. Zhang and colleagues [77] further described that “garlic oil and black pepper oil showed the highest activity against *B. duncani* growth”. According to Salama and colleagues [97], in an in vivo murine model, parasitemia of *Babesia microti* significantly decreased in allicin-treated mice (*p* < 0.01) from days 4–14 post-inoculation, as compared to controls.

#### 3.1.3. *Andrographis paniculata*

While a recent in vitro study by Feng and colleagues [68] did not support the efficacy of *Andrographis paniculata* against *Borrelia* species, it does have previous research to support its efficacy against Leptospirosis, another spirochetal illness [98]. 

#### 3.1.4. *Artemisia annua*

Also called sweet wormwood, Chinese wormwood, and Qing Hao, *Artemesis annua* has been used for over 2000 years [99] with particular attention to treating malaria [100]. Active components of *Artemesia annua* include artemisinin [101], artesunate, and artemether, all of which are potent antimalarial drugs [102]. The active ingredient artemisinin has been shown to have in vitro activity against stationary phase *B. burgdorferi* persister cells [103,104]. In humans, *Artemisia annua* has been used safely in doses up to 2250 mg daily for up to 10 weeks, and 1800 mg daily has also been used safely for up to six months [105]. At higher doses, gastrointestinal upset has been reported, including mild nausea, vomiting (rarer), and abdominal pain. 

#### 3.1.5. *Berberis vulgaris*/Berberine

*Berberis vulgaris* (*B. vulgaris*), is naturally found in semi-tropical areas in Africa, Asia, Europe, North America, and South America. Pharmacologically, *B. vulgaris* has been shown to have numerous properties, including anti-inflammatory, sedative, antipyretic, antiemetic, antioxidant, anti-cholinergic, anti-arrhythmic, and antimicrobial (including anti-malarial) properties [106,107]. Two active ingredients, berberine and berbamine, further demonstrate hypoglycemic, anti-inflammatory, hypotensive, antioxidant, and hypolipidemic properties [108,109]. Antimicrobial effects have been reported against multiple pathogens, including *Trichomonas vaginalis*, *Giardia lamblia*, *Entamoeba histolytica*, and certain *Leishmania* strains. According to Subecki and colleagues [110], berberine was shown to have inhibitory effects against *Babesia gibsoni*. According to Batiha and colleagues [111], a methanolic extract of *B. vulgaris* (MEBV) restricted the multiplication of several *Babesia* species, including *B. bovis*, *B. bigemina*, *B. divergens*, and *B. cabali*. Elkhateeb and colleagues [112] reviewed the anti-babesial activity of nine North African medicinal plants against the growth of *Babesia gibsoni* in vitro. Extracts prepared from *Berberis vulgaris* and *Rosa damascene* showed more than 90% inhibition at a concentration of 100 mcg/mL. Multiple isolates were shown to be active against *B. vulgaris*, with the most potent being syringic acid. Finally, berberine has been shown by Li and colleagues [84] to have relatively high activity against stationary phase *Bartonella henselae*.

#### 3.1.6. *Cinnamomum* (Cinnamon)

Cinnamon is a tropical Asian spice and a native plant of Sri Lanka and is extracted from the *Cinnamomum* genus, including *Cinnamomum camphora*, *Cinnamomum osmophloeum*, *Cinnamomum burmannii*, *Cinnamomum zeylanicum*, *Cinnamomum cassia*, and *Cinnamomum verum* [113]. Cinnamaldehyde, a key active ingredient, has been shown to have efficacy against the active spirochetal [67,96] and stationary forms of *Borrelia burgdorferi sensu latu* [67,73,96], as well as to the bio-film forms [67]. Feng and colleagues [73] found that essential oil from cinnamon bark (at a low concentration of 0.25%) was one of five essential oils with high anti-persister activity and better activity than the known persister drug daptomycin. Further, Feng and colleagues [73] reported that cinnamon bark had anti-*Borrelia* biofilm activity.

With regard to non-*Borrelial* tick borne pathogens, Ma and colleagues [82] reported that cinnamon bark and cinnamon leaf were both active against the stationary phase non-growing *B. henselae*, and also had good activity against log-phase growing *B. henselae*. Batiha and colleagues [76] provided evidence that several *Babesia* strains were inhibited in vitro by derivatives of cinnamon including *B. bovis*, *B. bigemina*, *B. divergens*, *B. caballi*, and *T. equi*. Furthermore, this study also showed that cinnamon was effective in vivo against *Babesia microti*.

#### 3.1.7. *Cistus creticus*

In vitro studies have shown that *Cistus* species have antibacterial, antifungal, antiviral, and anti-inflammatory properties [114,115,116,117,118,119,120,121,122]. In one study, volatile oil from *Cistus creticus* was shown to inhibit growing forms of *B. burgdorferi*. This study also found that essential oil volatile compounds exhibited a stronger growth inhibitory effect than leaf extracts [123]. Feng and colleagues [68,73] reported that Carvacrol was likely the active *Cistus creticus* component against log and stationary phase *Borrelia burgdorferi*.

#### 3.1.8. *Cryptolepis sanguinolenta*

Pharmacologic properties of *Cryptolepis sanguinolenta* have been shown to include anti-inflammatory, antimicrobial, anti-amoebic, anti-cancer, and anti-malarial features [124,125,126,127,128]. Cryptolepine, an active ingredient of *Cryptolepis sanguinolenta*, has been shown in vitro to have significant anti-plasmodial activity against drug-sensitive and drug resistant malaria strains [129,130]. 

Further, two open label trials using different formulations of *Cryptolepis sanguinolenta* have provided both efficacy and safety in the setting of treating patients with uncomplicated malaria [131,132]. Zhang and colleagues [74] provided in vitro evidence that *C. sanguinolenta* had efficacy against *Borrelia burgdorferi*, *Bartonella henselae*, and *Babesia duncani*, with good efficacy against both growing and stationary forms. Ma and colleagues [133] also identified efficacy against *Bartonella* stationary and active log phase forms. Of multiple agents studied, *cryptolepine* had the strongest inhibitory activity against *Babesia duncani*, including traditional combinations of quinine and clindamycin. *Cryptolepis sanguinolenta* has been shown to be safe at doses under 500 mg/kg of body weight [129]. However, the question regarding potential anti-fertility and embryotoxic effects [134,135] needs further study. 

#### 3.1.9. *Dipsacus sylvestris*/*Dipsacus fullonum* (Teasel Root)

*Dipsacus* is a species of plant native to Eurasia and North Africa, otherwise known as wild teasel or fuller’s teasel. Leopold and colleagues [136] studied several preparations of this agent against *Borrelia burgdorferi sensu strictu.* The hydroethanolic extract showed no growth inhibition, whereas significant growth-inhibiting activity could be shown in the two less polar fractions. Liebold and colleagues [136] did not study the efficacy of teasel root extract against round body forms or biofilm. However, Goc and Roth [137] did, finding no impact on persister forms of *Borrelia*.

#### 3.1.10. *Eugenia caryophyllata* (*Syzigium aromaticum* L. (Myrtaceae)

*Eugenia caryophyllata*, also known as clove, is a dried flower bud belonging to the Myrtaceae family that is indigenous to the Maluku islands in Indonesia [138]. Several important active components have been isolated in clover oil, including eugenyl acetate, eugenol, and β-caryophyllene. Many studies have examined *S. aromaticum* activity against various pathogenic parasites and microorganisms, including *Plasmodium*, *Babesia*, *Theileria* parasites, Herpes simplex, and hepatitis C viruses [138]. Feng and colleagues [73] identified that clove represented one of the top five essential oils studied (oregano, cinnamon bark, clove bud, citronella, and wintergreen) effective at low concentrations against the *B. burgdorferi* persister cell stationary phase. In fact, these five agents had greater activity than the known anti-persister drug daptomycin. Further, these researchers identified that clove, and cinnamon bark and oregano, were the top three essential oils with excellent *Borrelia* anti-biofilm effects. In addition, Ma, and colleagues [82] identified clove bud as one of 32 essential oils that had high activity against both the stationary non-growing phase and the log-phase growing forms of *Bartonella henselae*. Finally, in an in vivo murine model, Batiha and colleagues [139] published a study supporting the clove extract inhibition of *Babesia microti*.

#### 3.1.11. Grapefruit Seed Extract (GSE)

Goc and Rath [137] showed that GSE had in vitro activity against both stationary and active forms of *B. burgdorferi*. However, these positive findings contrasted with negative findings by Feng and colleagues [68], who opined that this disparity could have been explained by different GSE formulations, toxins present in the GSE formulations themselves, and/or different *Borrelia* strains tested. 

#### 3.1.12. *Juglans nigra* (Black Walnut)

There is evidence that *Juglans nigra* (black walnut) and its constituents have broad therapeutic properties, including antioxidant, antibacterial, antitumor and chemoprotective effects [140,141]. Regarding potential impacts on *Borrelia burgdorferi* and *Borrelia garinii*, *Juglans nigra* has exhibited bacteriostatic activity against log phase spirochetes of *B. burgdorferi* and *B. garinii* and bactericidal activity against *Borrelia* round bodies [137]. Ma and colleagues [133] also identified efficacy against stationary and active log phase forms of *Bartonella*. Although reports of gastrointestinal disturbance [142] and changes in skin pigmentation have been published [143,144], *Juglans nigra* is generally well-tolerated and side effects are uncommon. Reports in humans suggest that there may be some allergic cross reactivity in those allergic to tree nuts or walnuts, as well as cases of dermatitis [145]. 

#### 3.1.13. Monolaurin

Monolaurin first became available as a nutritional formulation in the mid-1960s and today is sold worldwide as a dietary supplement, commonly known by one of its chemical names, glycerol monolaurate (GML). The richest dietary source is coconut oil [146]. Monolaurin has been shown to have antibacterial activity against several Gram-positive strains such as *Staphylococcus* sp., *Corynebacterium* sp., *Bacillus* sp., *Listeria* sp., and *Streptococcus* sp. [147]. Goc and colleagues [66] found that baicalein and monolaurin were effective against biofilm-like colonies of both *B. burgdorferi* and *B. garinii*. 

#### 3.1.14. *Nigella sativa* (Black Cumin) 

*Nigella sativa* is known for its antioxidant, anti-inflammatory, antibacterial, antiviral, antiparasitic, anticarcinogenic, antiallergic, antidysenteric, and antiulcer effects. The main component within *N. sativa* is thymoquinone, which has been proven in vitro and in vivo against the growth of the malarial agent *Plasmodium berghei* in mice. El-Sayed revealed in vivo the efficacy of thymoquinone against *Babesia microti* in a murine model [148].

#### 3.1.15. Oregano

The herb oregano comes from the dried leaves of *Origanum* sp. (Mediterranean variety) or *Lippia* sp. (Mexican variety) [149]. Oregano is widely used in foods and has been designated GRAS (generally recognized as safe) by the FDA. In a small study, 200 mg/day emulsified *O. vulgare* oil was administered for 6 weeks. Carvacrol has been shown to be the most active component of oregano oil, and in one study showed excellent activity against *B. burgdorferi* stationary phase cells. This same study showed that other oregano oil compounds, p-cymene and α-terpinene, had no apparent activity. In addition, Feng and colleagues [73] found that oregano was efficacious against *Borrelia* biofilm microcolonies [73], and that oregano was one of the top five essential oils tested for activity against *B. burgdorferi* stationary phase culture. Further, these researchers found that carvacrol had efficacy against log phase *Borrelia burgdorferi* [73]. Ma and colleagues [82] reported that oregano was active against the stationary phase of non-growing *B. henselae* and had good activity against log-phase growing *B. hensela*. Of note, oregano has been found to have good blood–brain barrier penetration [150].

#### 3.1.16. *Otoba parvifolia* (Banderol)

According to Goc and Rath, *Otoba parvifolia* has been shown to have in vitro efficacy on active and dormant forms of *Borrelia burgdorferi sensu stricto.* Further, particularly in combination with *Uncaria tomentosa*, these agents have provided significant effects on all morphological forms of *B. burgdorferi* [137].

#### 3.1.17. *Piper nigrum* (Black Pepper)

As previously mentioned, according to Zhang and colleagues [77], “garlic oil and black pepper oil showed the highest activity against *B. duncani* growth”. 

#### 3.1.18. *Polygonum cuspidatum* (Japanese Knotweed)

Resveratrol represents one of the main active ingredients in *P. cuspidatum.* This agent has documented anti-tumor, antimicrobial, anti-biofilm, anti-inflammatory, neuroprotective, and cardioprotective effects [151,152,153,154,155,156]. Further, *P. cuspidatum* showed strong activity against both growing and non-growing stationary phases of *B. burgdorferi* [68]. According to Zhang and colleagues [74], *P. cuspidatum* showed a good inhibitory effect against *Babesia duncani* and *Bartonella henselae*. Ma and colleagues [133] also identified *P. cuspidatum* efficacy against *Bartonella* stationary and active log phase forms [133]. *P. cuspidatum* has been found to have minimal toxicity in animal and human studies. Gastrointestinal upset and diarrhea can occur, but generally resolve after decreasing or stopping the intake [157]. 

#### 3.1.19. *Rhus coriaria* L. (Sumac)

*Rhus coriaria* L. (Anacardiaceae) is commonly known as sumac. The most conventional use is as a spice, condiment, and flavoring agent, especially in the Mediterranean region. Accumulating evidence supports the antibacterial, antinociceptive, antidiabetic, cardioprotective, neuroprotective, and anticancer effects of this plant. Toxicity studies show that sumac is generally very safe to consume by humans, with little toxicity [158]. Batiha and colleagues [111] studied an acetone extract of *R. coriaria* (AERC), which was shown to have an inhibitory effect on all *Babesia* strains tested; AERC suppressed *B. bovis*, *B. bigemina*, *B. divergens*, and *B. caballi*.

#### 3.1.20. Rosmarinic Acid

Rosmarinic acid is an active component of the Lamiaceae (Labiatae) family and has been associated with anti-inflammatory and antioxidant activity [159]. According to Goc and colleagues [160], the addition of rosmarinic acid in conjunction with either luteolin or baicalein had additive bacteriostatic effects on *B. burgdorferi* anti-spirochetal activity.

#### 3.1.21. *Scuttelaria* spp., Baicalin, and Baicalein 

Extracted from the roots of *Scutellaria baicalensis* and *Scutellaria lateriflora*, the active flavonoid baicalin is a flavone glycoside obtained via the binding of glucuronic acid.

Another flavonoid, baicalin, has been shown to have anti-biofilm properties in other infections [161] and can work synergistically with other antimicrobials [162,163,164]. Specifically relating to tick-borne infections, Goc and colleagues [66] provided in vitro evidence that baicalein exhibited activity against the three known morphologic forms of *B. burgdorferi* and *B. garinii*. These forms included log phase spirochetes, latent round bodies, and biofilm. Feng and colleagues [68] independently provided evidence that this agent was effective in inhibiting both the active and stationary forms of *Borrelia burgdorferi*. In their 2021 paper, Zhang and colleagues [74] provided evidence that baicalein showed good in vitro activity against *B. duncani*, superior to both quinine and clindamycin. Interestingly, Goc and colleagues [160] have been able to show synergistic outcomes when baicalein was combined with other nutraceuticals, such as luteolin. This combination showed synergistic cooperation in killing knob-/round-shaped persistent forms and additive effects in the eradication of biofilms formed by the *Borrelia burgdorferi* and *B. garinii* studied. This combination was able to eliminate ~90% of active and persistent forms, as well as eradicating 50% of the mature *Borrelia* biofilms. Furthermore, this synergism was able to extend to formal antimicrobials, such as doxycycline. This combination was synergistic in treating all forms of *Borrelia*, including active spirochetal, stationary, and biofilm persisters.

*Scutellaria baicalensis* has documented clinical safety [165,166] and baicalein even exhibits a hepato-protective effect in preventing acetaminophen-induced liver injury [167]. There have been reports of sedation associated with activity on GABA receptor sites [168].

#### 3.1.22. *Stevia rebaudiana*

Certain formulations of stevia have been shown to be effective against the active, stationary, and biofilm forms of *B. burgdorferi* [137]. However, there were several formulations derived from stevia that researchers did not find to be effective. The lack of efficacy was corroborated by Feng and colleagues [68]. It is possible that activity may be preparation-specific, which could explain these divergent findings. However, additional research is needed to further understand the current mixed outcomes. Theophilus and colleagues [169] also showed that when stevia was used with three different antibiotics (doxycycline, cefoperazone, and daptomycin) there was a significant reduction in *B. burgdorferi* biofilm forms. Safety has been demonstrated at high levels of dietary intake [170,171].

#### 3.1.23. *Uncaria tomentosa* (Cat’s Claw)

There is evidence that *Uncaria tomentosa* has antimicrobial effects against human oral pathogens [172]. Research by Feng and colleagues [68] showed in vitro activity against stationary forms of *B. burgdorferi*, but less impact against the actively growing spirochete form. However, according to Goc and Rath [137], *Uncaria tomentosa* was shown to be effective against all morphological forms of *Borrelia burgdorferi sensu stricto.* Published safety data have shown minimal side effects in both animal and human models [173]. Human studies reflected a side effect profile comparable to a placebo in studies ranging from four weeks [174] to 52 weeks [175]. There are reports that *Uncaria* can have an impact on estrogen binding [176], with potential implications on contraceptive effects [177].

#### 3.1.24. Vitamin C

One peer-reviewed publication described the efficacy of vitamin C against the active, spirochetal forms of *Borrelia burgdorferi* (and to a lesser extent *Borrelia garinii)* in vitro [66]. This was supported by another study, which showed additive effects of vitamin C combined with doxycycline against the spirochetal form [178].

Of note, in clinical practice it is recommended that Vitamin C be avoided in combination with hydroxychloroquine due to a possible decrease in hydroxychloroquine efficacy [179].

Furthermore, if supraphysiologic doses of vitamin C are used it is important to rule out G6PD enzyme deficiency due to the risk of potentially triggering hemolysis.

Rizk and colleagues [180] showed that Vitamin C in conjunction with anti-*Babesia* agent diminazene aceturate (DA) improved the inhibition of *Babesia microti* growth in vivo. This was initially evaluated in an in vitro model using *Babesia bovis* and subsequently in an in vivo murine model using *Babesia microti*. The authors propose this combination as a viable option to decrease drug resistance.

#### 3.1.25. Vitamin D3

The immune system has been established as an important target of vitamin D. The active form of vitamin D (1,25-Dihydroxycholecalciferol [1,25-(OH)2D3]) directly and indirectly suppresses the function of pathogenic T cells while inducing several regulatory T cells that suppress multiple sclerosis and inflammatory bowel disease development [181]. Cantorna and colleagues [182] induced a murine arthritic condition through the injection of collagen or *B. burgdorferi*. Vitamin D3 supplementation dramatically decreased mouse arthritic symptoms (inflamed and swollen ankles and paws).

According to Goc and colleagues [66,73], Vitamin D has efficacy against the active spirochetal form of *Borrelia burgdorferi* but not the stationary or biofilm forms [66].

### 3.2. Combination Protocols Reveal Synergy

In 2017, Goc and colleagues [160] published data on several combinations that showed synergy in vitro against 3 forms of *Borrelia* (*garinii* and *burgdorferi*): active spirochetal, and both stationary and biofilm persister forms. These combinations are reflected in Table 2.

In 2020, Goc and colleagues [183] demonstrated the efficacy and synergy of combination treatment in a Lyme disease animal model and human volunteers. In this study, the researchers used baicalein, luteolin, and rosmarinic acid, along with monolaurin, cis-2-decenoic acid, and iodine/kelp. Inflammatory cytokines such as IL-6, IL-17, TNF-α, and INF-γ were elevated in infected animals, supporting active infection. These inflammatory cytokines normalized in infected animals that were subsequently treated. Furthermore, four weeks of dietary intake of this composition reduced the spirochete burden in animal tissues by about 75%. In a human study of 17 volunteers administered this composition, 67.4% of participants with clinical late or persistent LD (who had not responded to previous antibiotic treatment) responded positively with improved energy status, as well as improved physical and psychological well-being. Additionally, 17.7% had slight improvement, and 17.7% were nonresponsive. In the human observational study, patients received the treatment intervention in the form of capsules containing baicalein 250 mg/day, luteolin 75 mg/day, rosmarinic acid 100 mg/day, monolaurin 250 mg/day, 10-HAD 100 mg/day, and iodine 0.15 mg/day (in the form of kelp) three times per day for six months. These doses represented 1/8th of the minimum inhibitory concentration (MIC) values [183].

## 4. Miscellaneous

### 4.1. Synthetic Products

#### 4.1.1. Methylene Blue

Methylene blue (MB) has been used medically for centuries and is best known as an antidote for acquired methemoglobinemia (MetHB) [184]. Methylene blue was the first synthetic antimalarial to be discovered [185], and other common uses include septic shock [186] and as an intraoperative dye [187]. According to Feng and colleagues [104], MB has significant benefits in treating *Borrelia burgdorferi* stationary persister cells. In addition, Ma and colleagues [82] provided evidence of activity against stationary forms of Bartonella, and Li and colleagues [84] showed that MB had significant effects on the stationary and active log phase forms of *Bartonella*. Zheng and colleagues [85] reported that MB had relatively low MICs against the log phase form of *Bartonella*, and was more active than the doxycycline/gentamicin combination against the stationary forms. Lastly, MB and rifampin were the most active agents against the biofilm *B. henselae* after 6 days of drug exposure. Given its original use as an antimalarial and similarities in pathophysiology and treatment response between Malaria and *Babesia* [188,189], one could extrapolate that this agent should have anti-*Babesia* properties. Carvalho and colleagues [190] studied six agents combined with Artusenate in vitro against *Babesia bovis* and found that the most active drug was MB.

#### 4.1.2. Tetraethylthiuram Disulfide (Disulfiram)

Disulfiram has historically been used as an adjunct to sobriety maintenance in alcoholics. As a repurposed therapy, it has been found to be highly effective as an antiparasitic [191], antifungal [192], and anti-MRSA agent [193], along with having anti-mycobacterium tuberculosis activity [194]. Disulfiram has also been shown to be therapeutic against malaria [195]. Given the clinical and pathophysiological similarities between malaria and the tick-borne pathogen *Babesia* [188,189], one could propose potential efficacy for *Babesia* as well.

According to Potula and colleagues [196], disulfiram has excellent borreliacidal activity against both the log and stationary phase *B. burgdorferi sensu stricto B31 MI.* Their in vivo murine studies showed that disulfiram eliminated the *B. burgdorferi sensu stricto B31 MI* completely from the hearts and urinary bladder by day 28 post infection. This was also associated with reduced lymphadenopathy and inflammatory markers [196]. In fact, disulfiram was shown to be the most potent of 4366 compounds tested in vitro, with anti *B.burgdorferi* persister cell activity [197].

In a case series of three patients with *Babesia*, Liegner [198] was able to show a significant clinical benefit with the use of disulfiram. In a subsequent human study of 71 patients with suspected Lyme disease, 62 of 67 (92.5%) patients treated with disulfiram endorsed a net benefit on their symptoms. Furthermore, 12 of 33 (36.4%) patients who completed one or two courses of “high-dose” therapy (≥4 mg/kg/day vs. a “low-dose” group < 4 mg/kg/day) achieved an “enduring remission”, defined as remaining clinically well for ≥6 months without further anti-infective treatment. Of note, higher doses did produce a higher incidence of adverse reactions including fatigue (66.7%), psychiatric symptoms (48.5%), peripheral neuropathy (27.3%), and the mild to moderate elevation of liver enzymes (15.2%) [199]. To date, there is no evidence in the literature for a therapeutic effect of disulfiram on any *Bartonella* species.

### 4.2. Safety

According to Di Lorenzo and colleagues [200], the risk of side effects from botanical medicines is generally felt to be rare. However, each botanical medicine has its own unique action and physiological effect. As with any intervention, there are potentially unforeseen drug interactions or independent idiosyncratic responses [201]. As with any medical therapy, it is recommended that patients work with a knowledgeable practitioner who can advise and guide on the optimal and safest course of therapy. The potential for adverse responses should not preclude the value of the appropriate and judicious use of nutraceuticals, but rather emphasize the need for appropriate oversight. Foundational knowledge of potential interactions and side effects, as well as the use of surveillance labs to identify early adverse hepatic, renal, or bone marrow responses, is recommended.

### 4.3. Potential Criticisms

This paper attempted to organize the data in a format most useful for decision making at the point of care. However, it must be pointed out that there was heterogeneity in the diagnostic systems identifying pathogens, as well as in the source and potential quality of nutraceutical products being tested. In addition, studies on different pathogen strains were grouped together, and it must be recognized that different strains may respond differently to a given interventions or treatment. For example, with regard to *Borrelia*, some studies described *B. burgdorferi* and/or *B. garinii* strains. Studies on *Babesia* generally focused on *B. microti*, but occasionally *B. duncani*, and in some studies animal strains such as *B. gibsoni* were evaluated. In addition, most of the conclusions described were derived from in vitro or animal data.

In vitro studies can help assess the potential impact a therapy may have on a pathogen as a direct antimicrobial. However, in vitro studies exist outside the biological organism and thus cannot assess many of the broad pathways through which botanical medicines and nutraceuticals exert their effects. These include anti-inflammatory/anti-cytokine activity, immune system regulation/augmentation, the adaptogenic stimulation of cellular and organismal defense systems, and biofilm disruption, to name a few. Performing in vivo and particularly human research is vital as a future consideration. This would include larger studies evaluating specific interventions and protocols for specific pathogens. Further safety analysis of individual and combined regimens, along with potential drug interactions of both prescriptive and over-the-counter agents, would be of value.

## 5. Conclusions

In this paper, we reviewed the literature for the broader treatment of tick-borne infections, including *Borrelia*, *Bartonella*, and *Babesia*. By embracing the utility of botanical medicines and nutraceutical agents, our hope is that the field can decrease the need for long-term prescription antibiotics and antimicrobials, while improving overall outcomes. Furthermore, this broader treatment toolkit takes into account the multiple interconnected pathways through which these complex persistent infectious organisms affect the human organism. Ideally, prospective studies will allow for the further expansion of these therapeutic options. As further evidence is generated to support the safety and utility of the interventions discussed, we hope that practitioners can incorporate many of these concepts into their therapeutic strategies. It is our intention to empower clinicians with expanding options, while improving the quality of life and outcomes of our patients.

## Figures and Tables

**Table 1 microorganisms-11-01759-t001:** Research by Goc and colleagues [67]; analyzed organic oil for activity against 3 forms of *Borrelia* spp.: active, stationary, and biofilm persister forms.

Organic Oils	Active Ingredient	Active	Stationary	Biofilm
Bay leaf oil (*Pimenta racemosa*)	Eugenol	X	X	X
Birch (sweet) oil (*Betula lenta*)	Methyl salicylate	X	X	X
Cassia oil (*Cinnamomum cassia*)	Cinnamaldehyde	X	X	X
Chamomile oil German (*Matricaria chamomilla*)	Chamazulene	X	X	X
Thyme oil (*Thymus vulgaris*)	Thymol	X	X	X

**Table 2 microorganisms-11-01759-t002:** Summarizing various combinations of agents with synergistic or additive impact on spirochetal, stationary and biofilm forms against the two strains of *Borrelia* studied.

Synergistic or Additive Combinations	Spirochete	Stationary	Biofilm
Baicalein with luteolin	X	X	X
Monolaurin with cis-2-decenoic acid	X		X
Baicalein and rosmarinic acid	X		
Luteolin and rosmarinic acid	X		
Baicalein and iodine		X	
Luteolin and iodine		X	
Baicalein with cis-2-decenoic acid	X		
Luteolin with cis-2-decenoic acid	X		

## Data Availability

Not applicable.

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
