# Peer review of "The Use of Natural Bioactive Nutraceuticals in the Management of Tick-Borne Illnesses"

_microorganisms, 2023, doi:10.3390/microorganisms11071759_

Round 1

Reviewer 1 Report

Comments for authors

An interesting paper. In general article is well written. The data and the meta-analysis are certainly valuable and will add to the knowledge of scientific community. The analysed problems are very important. But editing and wording of the text should be carefully checked and corrected in many places: punctuation, spaces, italics, nomenclature. The edition is generally untidy.

Summarizing, the data presented in reviewed manuscript are certainly worth publishing, but some corrections are recommended as below:

  1. In my opinion the publications 203 and 204 from references are not cited in the manuscript. Therefore, the citation of references in the manuscript should be checked and eventually corrected. All publications listed in references should be cited in text of the manuscript.
  2. Each figure and/or each table should be clear separately, independently of the text and another tables or figures. Therefore titles of tables and figures should be corrected.
  3. In my opinion, spaces and the punctuations should be carefully checked and corrected.
  4. Generally,  the citation of references in the manuscript should be carefully checked and eventually corrected.

Author Response

1.    In my opinion the publications 203 and 204 from references are not cited in the manuscript. Therefore, the citation of references in the manuscript should be checked and eventually corrected. All publications listed in references should be cited in text of the manuscript.
Thank you for pointing out this discrepancy. I have corrected the organizational error
2.    Each figure and/or each table should be clear separately, independently of the text and another tables or figures. Therefore titles of tables and figures should be corrected.
addressed
3.    In my opinion, spaces and the punctuations should be carefully checked and corrected.
addressed
4.    Generally,  the citation of references in the manuscript should be carefully checked and eventually corrected.
addressed

Reviewer 2 Report

1) The methods applied in the research are incomplete and/or flawed, since it does not express the inclusion and exclusion criteria, and uses only a database.

2) If 400 references were used, why are only 204 listed? How was this selection made?

3) In several parts of the text the term, in Latin, “in vitro” must be placed in italics

4)Discussion

Table 1 - there is no mention of table 1 inserted on line 111.

In several parts of the text the term, in Latin, “in vitro” must be placed in italics

Line 118 - insert italics in Uncaria tomentosa

Line 130 to 134 - “Zhang et al., some botanical medicines and their active constituents have potent activity against B. duncani in vitro and may be further explored for more effective treatment of babesiosis. In one study, a panel of herbal medicines were screened and identified that Cryptolepis sanguinolenta, Artemisia annua, Scutellaria baicalensis, Alchornea cordifolia, and Polygonum cuspidatum had good in vitro inhibitory activity against B. duncani in a hamster erythrocyte model”

I suggest indicating the route of administration, including the other experiments cited as well.

Line 142 and 261 - insert italics in Bartonella

Line 179 and 351 - remove Babesia's “dot”. duncani

Lines 188 and 190 - insert italics in “B. burgdorferi” and “in vivo”, respectively

Line 313 - Staphylococcus sp., remove italics from sp.

Line 314 - Streptococcus sp [148]., insert dot in sp.

Standardize the abbreviation of B. burgdorferi, Babesia duncani, Bartonella henselae and among others, as mentioned at length earlier

Line 319 - insert italic N. sativa

Line 386 - studied Borrelia burgdorferi and garinii - insert B. garinii

Line 426 - “Rizk et al. showed that Vitamin C in conjunction with anti-Babesia agent diminazene aceturate (DA) improved the efficacy of DA”

In this case, I suggest inferring in which situation did it improve efficacy? Reducing the dose?

Inha 436 - “According to Goc et al. [ref] Vitamin D has efficacy against the active spirochetal form of Borrelia burgdorferi but not the stationary or biofilm forms” - in what form?

Table 2 - format Bgarinii

5) In the final considerations, it is important to emphasize that bioactives and nutraceuticals are components of a adjuvant treatment, and that more in vivo research should be carried out, assessing the ideal dose for the best response of the animal, if there is a toxicity effect, among other information.

Author Response

1)    The methods applied in the research are incomplete and/or flawed, since it does not express the inclusion and exclusion criteria, and uses only a database.
A more robust description is included:
Methods
Using PUB MED, comprehensive searches were employed, including using the following search terms “Lyme disease,” “Borrelia,” “Bartonella,” “Babesia,” and cross referencing with generic concepts such as “herbals” “nutraceuticals” “botanical medicines” along with each of the multiple agents described in this manuscript. Over 400 references were identified and reviewed. To meet inclusion criteria, studies had to include agents shown to have a therapeutic effect on at least one of the following: Borrelia burgdorferi and/or Bartonella species active, stationary or biofilm forms, or activity against Babesia strains. Papers not demonstrating treatment evidence against any of these pathogens were excluded and thus not referenced.
2)    If 400 references were used, why are only 204 listed? How was this selection made?
See methods section above
3)    In several parts of the text the term, in Latin, “in vitro” must be placed in italics
done
4)Discussion 
Addressed below in content of the manuscript
Table 1 - there is no mention of table 1 inserted on line 111.  
In several parts of the text the term, in Latin, “in vitro” must be placed in italics
Line 118 - insert italics in Uncaria tomentosa
Line 130 to 134 - “Zhang et al., some botanical medicines and their active constituents have potent activity against B. duncani in vitro and may be further explored for more effective treatment of babesiosis. In one study, a panel of herbal medicines were screened and identified that Cryptolepis sanguinolenta, Artemisia annua, Scutellaria baicalensis, Alchornea cordifolia, and Polygonum cuspidatum had good in vitro inhibitory activity against B. duncani in a hamster erythrocyte model”
I suggest indicating the route of administration, including the other experiments cited as well.
Line 142 and 261 - insert italics in Bartonella
Line 179 and 351 - remove Babesia's “dot”. duncani
Lines 188 and 190 - insert italics in “B. burgdorferi” and “in vivo”, respectively
Line 313 - Staphylococcus sp., remove italics from sp.
Line 314 - Streptococcus sp [148]., insert dot in sp.
Standardize the abbreviation of B. burgdorferi, Babesia duncani, Bartonella henselae and among others, as mentioned at length earlier
Line 319 - insert italic N. sativa
Line 386 - studied Borrelia burgdorferi and garinii - insert B. garinii
Line 426 - “Rizk et al. showed that Vitamin C in conjunction with anti-Babesia agent diminazene aceturate (DA) improved the efficacy of DA”  
In essence, outcomes improved
In this case, I suggest inferring in which situation did it improve efficacy? Reducing the dose?
Inha 436 - “According to Goc et al. [ref] Vitamin D has efficacy against the active spirochetal form of Borrelia burgdorferi but not the stationary or biofilm forms” - in what form? 
The active spirochetal form
Table 2 - format Bgarinii

5) In the final considerations, it is important to emphasize that bioactives and nutraceuticals are components of a adjuvant treatment, and that more in vivo research should be carried out, assessing the ideal dose for the best response of the animal, if there is a toxicity effect, among other information. 
4.3. Potential criticisms
This paper attempts to organize the data in a format most useful for decision making at the point of care. However, it must be pointed out that there is heterogeneity in the diagnostic systems identifying pathogens, as well as the source and potential quality of nutraceutical products being tested. In addition, studies on different pathogen strains were grouped together, and it must be recognized that different strains may respond differently to a given intervention or treatment. For example, with regards to Borrelia, some studies described B. burgdorferi and/or B. garinii strains. Studies on Babesia generally focused on B. microti but occasionally B. duncani, and in some studies, animal strains such as B. gibsoni were evaluated. In addition, most of the conclusions described are derived from in vitro or animal data.
In vitro studies can help assess the potential impact a therapy may have on a pathogen as a direct antimicrobial. However, in vitro studies exist outside the biological organism and thus cannot assess many of the broad pathways through which botanical medicines and nutraceuticals exert their effects. These include anti-inflammatory/anti-cytokine activity, immune system regulation/augmentation, adaptogenic stimulation of cellular, and organismal defense systems, and biofilm disruption to name a few. Performing in vivo and particularly human research is vital as a future consideration. This would include larger studies evaluating specific interventions and protocols for specific pathogens. Further safety analysis of individual and combined regimens, along with potential drug interactions of both prescriptive and over the counter agents would be of value.